# Optimisation on Thermoforming of Biodegradable Poly (Lactic Acid) (PLA) by Numerical Modelling

**DOI:** 10.3390/polym13040654

**Published:** 2021-02-22

**Authors:** Huidong Wei

**Affiliations:** 1College of Health and Life Sciences, Aston University, Birmingham B4 7ET, UK; h.wei1@aston.ac.uk; 2Rayner Intraocular Lenses Limited, Worthing BN14 8AQ, UK

**Keywords:** optimisation, thermoforming, PLA, numerical modelling, FEA, zones, biaxial strain

## Abstract

Poly (lactic acid) (PLA) has a broad perspective for manufacturing green thermoplastic products by thermoforming for its biodegradable properties. The mechanical behaviour of PLA has been demonstrated by its strong dependence on temperature and strain rate at biaxial deformation. A nonlinear viscoelastic model by the previous study was employed in a thermoforming process used for food packaging. An optimisation approach was developed by achieving the optimal temperature profile of specimens by defining multiple heating zones based on numerical modelling with finite element analysis (FEA). The forming process of a PLA product was illustrated by modelling results on shape evolution and biaxial strain history. The optimal temperature profile was suggested in scalloped zones to achieve more even thickness distribution. The sensitivity of the optimal results was addressed by checking the robustness under perturbation.

## 1. Introduction

Thermoforming is a widely used traditional processing technology in manufacturing variable plastic products for food packaging [1]. It evolves with time from experience-based to more scientific by the understanding on the thermomechanical behaviour of amorphous or semi-crystalline polymeric materials, leading to more accurate control of forming process and final products [2]. Poly (lactic acid) (PLA) is a bio-based polymer for the green food packaging, such as vegetable containers, cups and food trays that can be fully biodegradable [3,4,5]. Amorphous PLA had the glass transition temperature (T_g_) about 60 °C, cold thermal crystallisation temperature of 100 °C (T_cc_) and melting temperature (T_m_) of about 160 °C [6]. The processing temperature for thermoforming of PLA was between T_g_ and T_cc_ at semi-solid state of material to allow enough mobility of molecular chains but avoid crystallisation before forming. At this temperature range, the mechanical properties of PLA materials were characterised by its viscoelasticity at finite strain with strong dependence on temperature, strain rate and deformation mode. The increase of processing temperature weakened the stress response at equivalent strain level and significantly delayed the onset of strain hardening [7,8]. Stronger mechanical response and hardening behaviour was observed by improving the strain rate [9]. The influence from strain mode was revealed by applying controlled in-plane displacement along two orthogonal directions on thin films, i.e., equal biaxial (EB), constant width (CW) and sequential (SEQ) [10]. Compared with the CW deformation, there was higher stress response but smaller ultimate elongation at EB deformation along the stretching direction. The disadvantage of the previous study was constrained by the low-rate deformation (of less than 1 s^−1^) and at uniaxial strain [11]. In terms of the forming process of PLA above T_g_, it has been known that the biaxial deforming process was expected to happen at high-rate deformation (of between 1 and 17 s^−1^) by short forming time (of lower than 3 s) [12].

The thermoforming process of polymers was influenced by the selection of processing conditions, such as temperature, stretching speed and flow rate [13]. Among all of these factors, temperature had the most significant effect on the thermoformed products [14]. The temperature distribution can be controlled to build non-uniform conditions by zone-heating to prevent the thinner corners under uniform temperature profile [15,16]. The numerical approach by finite element analysis (FEA) offered a way to achieve process modelling of thermoforming validated by the experimental tests [17,18,19,20,21], which helped optimise the operation by virtual testing to improve the traditional trial-and-error method [17,18,19]. The temperature could be easily implemented into FEA modelling to perform process simulation at non-uniform or non-isothermal conditions [22,23,24]. One such application was on the optimisation of the temperature distribution to achieve desirable thickness distribution and/or the temperature sensitivity [22,23]. The key technical aspect for the process modelling relied on the correct expression of mechanical behaviour, i.e., constitutive model. The hyper-elastic model was the most widely used in thermoforming of polymers [22], whilst it was found that the rheological behaviour due to material viscosity significantly affected the thickness distribution after thermoforming [25]. To model the rate dependence of polymeric material above T_g_, the viscoelastic model was found to be more suitable for process modelling [26]. In contrast to the broad research on the processing of traditional polymers, the study on the thermoforming of PLA by numerical modelling was very poor.

It has been found that the processing condition of PLA significantly determined the thickness deviation, where a trial-and-error method was employed to find the optimal operating temperature [27]. The mechanical performance of PLA can be significantly improved by controlling the biaxial strain on the materials. At a low processing temperature (70 °C), the mechanical performance of stretched PLA after a biaxial strain showed a dramatic improvement of the Young’s modulus and elongation to break [28]. The morphology of materials can be enhanced by improving the strain rate and manipulating the mode of deformation [9,10]. All these studies suggest a need of optimising the thermoforming condition for the best performance of the products. Most constitutive models of PLA were developed at working temperature lower than T_g_ [29,30,31], which cannot suit the condition of the forming process. A recent study on mechanical behaviour of PLA at a wide range of temperatures (of between 70 and 100 °C) and strain rate (of between 1 and 16 s^−1^) provided a deep understanding on the thermomechanical behaviour of PLA between T_g_ and T_cc_ [32], and together with it, a nonlinear viscoelastic model was developed, which provided a chance of using numerical modelling to promote the thermoforming of PLA. By this motivation, a calibrated nonlinear viscoelastic model was employed to capture the deforming behaviour of PLA during thermoforming. An optimisation approach was developed by searching the optimal temperature profile of specimens based on numerical modelling and FEA. A thermoforming process of a PLA container was illustrated by modelling the shape evolution and biaxial strain history. The optimal temperature profile was suggested to achieve the even thickness distribution by more uniform biaxial stretch, whilst the sensitivity of the optimal results was addressed by examining the robustness under the perturbation of temperature.

## 2. Materials and Methods

The pellet-shaped material (Grade: PURAC LX175, Corbion, Amsterdam, The Netherlands) was supplied as a standard PLA suitable for film extrusion and thermoforming. The pellets were pre-dried at 80 °C for 12 h to remove moisture, as suggested by the product data sheet, and then extruded into sheets with the thickness of 1 mm by a single screw extruder. The extruded PLA sheet was cut into 76 × 76 mm square samples to test the thermomechanical behaviour under biaxial deformation. Two strain modes, i.e., equal biaxial deformation (EB) and constant-width (CW) deformation, were performed at different strain rates (1 to 16 s^−1^) at different processing conditions (from 70 to 100 °C). The overall experimental setup and the testing results have been introduced in the previous study [32].

A nonlinear viscoelastic model known as the glass-rubber model (GR) developed by Buckley was used to model the constitutive behaviour of amorphous polymers above T_g_ [33,34,35]. The model can be expressed as two Maxwell networks consisting of the bond-stretching stress to capture the initial elastic response and a conformational stress to simulate the strain hardening behaviour. Nonlinear viscosity was introduced in the two networks by Erying flow and the slippage transition to arrest, identified by the bonding viscosity (µ_0_) and conformation viscosity (γ_0_). The dependence of viscosity on the temperature was developed by two mathematical equations (Equations (1) and (2)), where the relationship between the viscosities (µ_0_, γ_0_) at processing temperature (T) and the viscosities (µ_0_^*^, γ_0_^*^) at reference temperature state (T^*^, T_s_^*^) was built. In the equations, two groups of material constants of the Conhen-Turbull constant (C_v_, C_s_) and the Vogel temperature (T_v_, T_v_^s^) together with the activation energy (H_0_) were calibrated. The material constants of the constitutive model in the previous study were employed [32].
(1)$$\ln\left(\frac{\mu_{0}}{\mu_{0}^{*}}\right)=\frac{C_{v}}{T-T_{v}}-\frac{C_{v}}{T^{*}-T_{v}}+\frac{H_{0}}{\mathrm{RT}}-\frac{H_{0}}{{\mathrm{RT}}^{*}}$$
(2)$$\ln\left(\frac{\gamma_{0}}{\gamma_{0}^{*}}\right)=\frac{C_{s}}{T-T_{v}^{s}}-\frac{C_{s}}{T_{s}^{*}-T_{v}^{s}}$$


A plug-assist thermoforming process for manufacturing a PLA container is shown in Figure 1. PLA materials were extruded into sheet with a thickness of 2 mm. The sheet was clamped on the frame of the mould and heated to the processing temperature (Figure 1a). The plug on top pushed the sheet down to the designed depth. The air trapped in the cavity of the mould was pumped to create a vacuum, thus compressing material onto the internal surface of the mould to form the desirable shape. The dimensions of the mould and plug are shown in Figure 1b.

Under axisymmetric loading and heating conditions, the axisymmetric deformation was expected in the thermoforming process. Based on the geometry of the setup, one fourth of the processing components (sheet, plug, mould) were built in a FEA model (Figure 1c). In the model, the plug and mould were defined as rigid parts (white in Figure 1c). The area with a distance of 5 mm to the edge on the PLA sheet for clamping was left and treated as a rigid zone (white in Figure 1c). The rest effective area had a radius of 60 mm, which was deformable (blue in Figure 1c). An initial distance of 5 mm between the plug and sheet was prescribed, where the plug was assigned a downward linear displacement of 64 mm at speed of 640 mm/s, i.e., duration of 0.1 s. Following the axial stretch, a positive pressure value of 6 bar (0.6 MPa) was applied on the top surface of the sheet to simulate the activation of internal vacuum at a duration of 0.1 s. Therefore, the total forming time was set to be 0.2 s. Two pairs of contact, one between the plug and sheet and the other between the mould and film, were established in the FEA model. The mesh had a precision of 2.5 mm and temperature values were defined on the nodes by the processing temperature.

Figure 2 displays that the surface of the PLA sheet was equally divided into a number of zones (N), i.e., N = 4 and N = 6, by two different division distances (15 and 10 mm). Each zone had a unique processing temperature (T_p_) to guarantee the axisymmetric deformation. The nodal temperatures in the FEA model were defined at a range from T_p_ = 72 to 77 °C and considered to be constant in a short duration of forming process (of 0.2 s). The temperature of the elements crossing the edge of two adjacent zones was interpolated by the nodal temperature.

An optimisation procedure was developed by a flowchart in Figure 3. A python script file (‘processOptimisation.py’) was used to control the whole optimising process, where an optimisation algorithm in an open-source library (‘Scipy: L-BFGS’) was employed. The initial processing temperature was set to be T_p_ = 77 °C for all the zones. An input file (‘thermoform.inp’) for the FEA model was built based on the geometric information, material properties and boundary/loading conditions aforementioned. This input file was updated by a python file (‘inpOperation.py’) using the new zone temperature values, which executed the FEA computation. When each computation cycle was completed, a result manipulation program (‘odbOperation.py’) read the resulted thickness and temperature of each zone from the output database (‘thermoform.odb’). These results were written into a recording file (‘record.dat’) to track the optimisation process. An objective function of minimal ratio was defined by assessing the ratio of biggest zone spatial thickness ((STH)_max_) over smallest zone spatial thickness ((STH)_min_). This ratio was optimised by updating the temperature profile until a minimal result was reached.

## 3. Results

### 3.1. Forming Process

At the initial prescribed processing temperature of T_p_ = 77 °C (Figure 4), thermoforming of a PLA container showed that the push from the axial motion of the plug produced an indented sheet (t = 0.05 s). As the further push was introduced, more materials started to build contact with the top end of the plug (t = 0.1 s). At this stage, no material was found to touch the mould. When the vacuum was applied, the sheet surface was inflated and contacted the side of mould primarily (t = 0.15 s). Then, the contact between the bottom surface and the mould was built after a further 0.05 s to form the final shape (t = 0.2 s).

The hoop strain (ε_11_) history (Figure 5a) displayed that the increase of strain occurred at zone 1 within a radius (R) of 15 mm before t = 0.1 s during the plug stretching process at a peak strain rate of 1.3 s^−1^. At this time period, the growth of axial strain (ε_22_) history at the 4 zones along the in-plane directions was illustrated at all the zones at different deformation stage (Figure 5b). The highest axial strain rate was found to be 26.6 s^−1^ at zone 1 and the lowest value was 3.1 s^−1^ at zone 4. At t = 0.1 s, the achieved biaxial strain at zone 1 was 0.3 (ε_11_) and 1.3 (ε_22_), respectively. There was constant width (CW) stretch at zones 2 to 4, with the axial strain (ε_22_) ranging from 0.2 to 0.5. After the activation of vacuum (t = 0.1 s), a dramatic increase of hoop strain (ε_11_) was observed along zones 1 and 2 within a radial distance (R) of 30 mm at a rate of 20.4 and 14.6 s^−1^ (Figure 5a). At the radius out of 30 mm, there was very low hoop strain rate at zone 3 (1.7 s^−1^) and zone 4 (0.4 s^−1^). Application of vacuum introduced a slight decrease of axial strain at zones 1 and 2 but an increase at zones 3 and 4 before t = 0.15 s (Figure 5b). After that, the axial strain increased again at zone 1 by a rate of 14.4 s^−1^ and zone 2 by a rate of 10.5 s^−1^, and a stable deformation was reached after t = 0.17 s at 4 zones. The overall forming process indicated a big stretch for regions near the centre (R < 30 mm) and a small deformation at other areas at uniform processing temperature profile (T_p_ = 77 °C).

### 3.2. Optimisation Result

Figure 6 shows the evolution of objective function at the optimisation process at two different divisions. At N = 4 (Figure 6a), there was a gradual increase of thickness ratio from 0.25 to 0.47 within 50 iterations. Beyond 50 iterations, an alternative evolution behaviour was observed during the optimisation process. At N = 6 (Figure 6b), it took about 90 iterations for the thickness ratio to increase from 0.30 to 0.52. Fluctuations were found as well at further iterations. This indicated that the optimisation process was sensitive to the temperature change at each step, which was set to an increment of 1 °C. The optimal result was selected at the peak level at the alternations and the temperature sensitivity would be further addressed by applying a perturbation value (of 1 °C) on the temperature profile.

The optimal temperature profile at each division case is shown in Figure 7. At N = 4 (Figure 7a), a low temperature level of T_p_ = 72 °C (zone 1) and 73 °C (zone 2) was found at R = 30 mm. At the region of zone 3 and 4, i.e., R = 30 to 60 mm, the optimal temperature was at the high level of T_p_ = 77 °C. At N = 6 (Figure 7b), a similar optimal temperature profile was observed, where there was a low temperature of T_p_ = 72 °C at zones 1 and 2 within R = 20 mm and high temperature of T_p_ = 77 °C at zones 4, 5 and 6 (R > 30 mm). A minor difference existed for N = 6 at zone 3 between R = 20 and 30 mm, with an optimal temperature of 75 °C, in comparison to the result of 73 °C at zone 2 (R = 15 to 30 mm) for N = 4.

By employing the optimal temperature (TEMP) profile at the two cases (N = 4 and 6), the spatial thickness (STH) distribution of formed sheet was compared with the result at the uniform temperature profile (T_p_ = 77 °C) in Figure 8. It can be seen that non-uniform temperature distribution was presented after optimisation at a similar gradient at the two cases, which introduced the change of thickness profile. For both cases, high temperature of T_p_ = 77 °C (red) covered the area near the edge of the sheet and there was a low-temperature region (T_p_ = 72 °C) (blue) near the central region. The temperature between T_p_ = 72 to 73 °C occupied the major transition part from low to high temperature at N = 4 (Figure 8a). This region had a high-level temperature of T_p_ = 73 to 75 °C at N = 6 (Figure 8b). Before optimisation, a thickness profile ranging from 0.4 to 1.6 mm along the whole sheet was found. By optimisation, the minimum thickness was improved to be over 0.61 mm. There was a similar thickness distribution of 1.02 to 1.65 mm (colour: light green, yellow and red) at both divisions (N = 4 and 6). The thickness ranging from 0.61 to 0.82 mm (colour: light blue) covered a smaller area at N = 4 than at N = 6. There was a wider distribution of thickness from 0.82 to 1.02 mm (colour: dark green) at the case of less division (N = 4).

In Figure 9, the thickness distribution along the radius from centre to edge illustrated a big increase from 0.5 to 0.7 mm by 40% within R = 50 mm after optimisation. A small decrease of thickness of 0.1 mm was observed from optimisation beyond R = 50 mm. The overall thickness ranged from 0.7 to 1.6 mm after optimisation. Between the distance of R = 30 to 50 mm, a smoother transition of thickness was exhibited at uniform temperature profile than the results at non-uniform cases. A step change of thickness was more evidently found at the optimised temperature at N = 4, indicating a sharp increase and decrease at distance from R = 30 to 60 mm. This tendency was weakened at N = 6 by introducing a small temperature gradient with more divisions.

The results of nominal strain in Figure 10a displayed a big decrease of hoop strain (ε_11_) at the radius of R = 30 mm from ε_11_ = 1.1 to 0.6 by an approximate decrease of 45%. The degree of decrease of hoop strain beyond R = 30 mm reduced gradually and disappeared after R = 72 mm. Figure 10b showed a decrease of axial strain (ε_22_) by optimisation from ε_22_ = 1.3 to 0.7 within a radius of R = 30 mm. No monotonic change of axial strain (ε_22_) at uniform temperature profile was observed compared to the result from optimisation. At N = 4, the decease of axial strain by optimisation continued until the radius of R = 48 mm and the elevation of axial strain occurred. This tendency showed some differences for the optimisation result at N = 6, indicating a dramatic decrease of axial strain and subsequent increase until R = 38 mm, after which the axial strain from optimisation exceeded the result at uniform temperature. Beyond R = 82 mm, there was a small difference of the hoop and axial strain before and after optimisation, leading to a slight drop of the wall thickness.

### 3.3. Temperature Perturbation

The thermal sensitivity of optimisation condition was studied by applying a temperature perturbation (of ±1 °C) on the optimal results of 2 cases (N = 4, 6) in Figure 11. At N = 4 (Figure 11a), a temperature difference was exhibited at low and high temperature levels. The thickness distribution showed the similar results, and a minimum thickness of 0.61 mm was found. The area with thickness ranging from 0.61 to 1.23 mm had similar distribution to the result without perturbation. At N = 6 (Figure 11b), an altering change of thickness was displayed within the radius of R = 20 mm by reducing the temperature. The overall thickness distribution had no difference after applying perturbation. The result of perturbation proved that the optimisation was primarily from the temperature profile at different zones based on the divisions with a tolerance of ±1 °C. At the optimal temperature distribution, enough robustness was illustrated, which helped develop the temperature-controlling strategy in the practice.

The thickness distribution at temperature perturbation is shown in Figure 12. At N = 4 (Figure 12a), a minor influence from temperature on the thickness was observed within the radius of R = 45 mm. The thickness moved to a slightly higher level by the decrease of temperature and a lower level due to the increase of temperature. There was no evident effect on the thickness beyond the radius of R = 45 mm. At N = 6 (Figure 12b), the temperature perturbation induced a similar trend with reduced thickness near the centre by increase of temperature within R = 35 mm. The result showed an improvement of 0.7 mm compared to the result of 0.5 mm before optimisation.

## 4. Discussion

Thermoforming of biodegradable PLA was investigated and optimised by numerical modelling together with finite element analysis (FEA). The highlight of the research was the application of a nonlinear viscoelastic model to capture the rate and temperature dependence of the material behaviour and an optimisation approach by combining FEA with general programming.

It has been found that the PLA materials at biaxial deformation showed a strong dependence on temperature and strain rate [32]. The importance of performing the biaxial stretching test was addressed by the deforming behaviour of PLA under thermoforming, implying a wide range of strain rate (of between 0.4 and 26.6 s^−1^) and unequal hoop and axial strain. This behaviour highlighted the suitability of using the viscoelastic model compared to the hyper-elastic model [26]. The PLA testing data used to calibrate the material constants of model was between 1 and 16 s^−1^ [32], which was proved to be incapable of covering the range of the strain rate encountered in the forming process. Despite the lack of enough fitting data, the model showed a reasonable extrapolating result in the process simulation. The reason behind this was the physically based mathematical expression of the glass-rubber model [33,34,35]. Other behaviour from the process modelling was the sequential hoop and axial deformation, which was observed by the stretch blow moulding of PLA products [12,36].

The deforming characteristics of plug-assist thermoforming of PLA showed similar duration and thickness distribution to the traditional material processing [14,19]. The strong influence from processing temperature of PLA materials enabled the optimising of thickness distribution by controlling the temperature profile, implying an optimal temperature profile by high-temperature region and the rest low-temperature region, the tendency of which was in accordance with the previous finding for plug-assist thermoforming of other polymers [23]. By applying the optimisation, the thickness distribution was significantly improved within a certain distance from the central axis, whilst the other regions were hardly affected. Even under the similar thickness, the biaxial stretch introduced in the forming process was found to be dependent on locations. The biaxial strain was primarily along the axial direction within 70% of the radial distance, where the optimal axial strain (of between 0.5 and 1.0) was higher than the maximum average strain (of between 0.2 and 0.5) in another study [27]. More numbers of divisions of temperature distribution introduced smoother transition of thickness but required a more accurate heating method [15,37], whilst the temperature control will not be so strict due to the minor effect from the thermal perturbation.

A narrow range of forming temperature (of between 72 and 77 °C) was studied despite a wide processing temperature window (of between 70 and 100 °C) of PLA [32,38]. The advantage of performing thermoforming at low temperature was the prevention of the cold thermal crystallisation during low-speed heating, whilst a more rapid heating can shift the cold crystallisation to occur at elevated temperature [11,39,40]. A significant mechanical enhancement could be achieved by the morphological rearrangement by applying a small biaxial stretch at low-temperature conditions [11,28,41]. The PLA material from thermoforming that experienced inhomogeneous biaxial stretch will lead to anisotropic mechanical performance of the post-stretch material. In addition to the influence from processing temperature and strain, the microstructure and mechanical properties of biaxially stretched PLA showed the dependence on the sequence of deformation and post-operation, such as annealing [10,36,39,42]. To provide an optimal mechanical performance of PLA products from thermoforming, more processing factors (wide-range temperatures, strain levels, stain paths, post-operation) together with the material properties after biaxial stretch need to be incorporated in the further study.

## Figures and Tables

**Figure 1 polymers-13-00654-f001:**
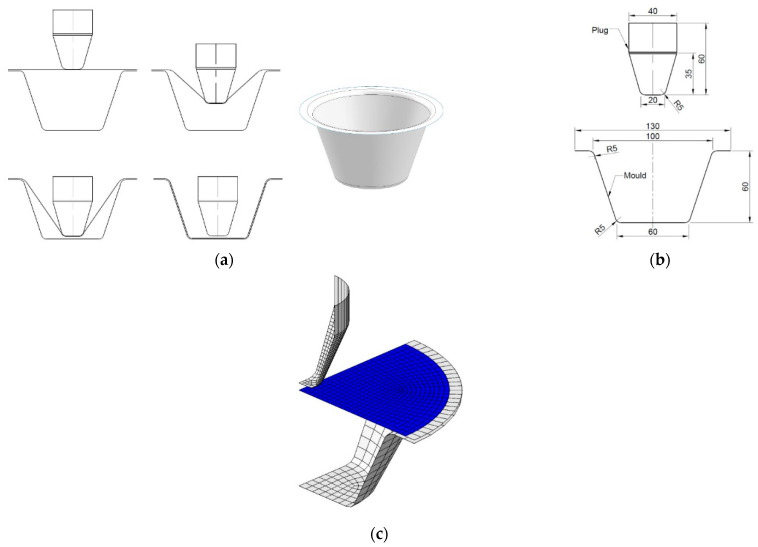
Plug-assist thermoforming. (**a**) Forming process, (**b**) design of plug and mould, (**c**) finite element model.

**Figure 2 polymers-13-00654-f002:**
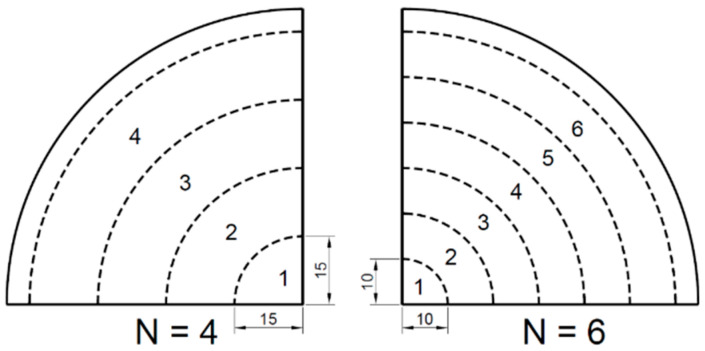
Zone divisions of poly (l-lactic acid) (PLA) sheets for thermal optimisation.

**Figure 3 polymers-13-00654-f003:**
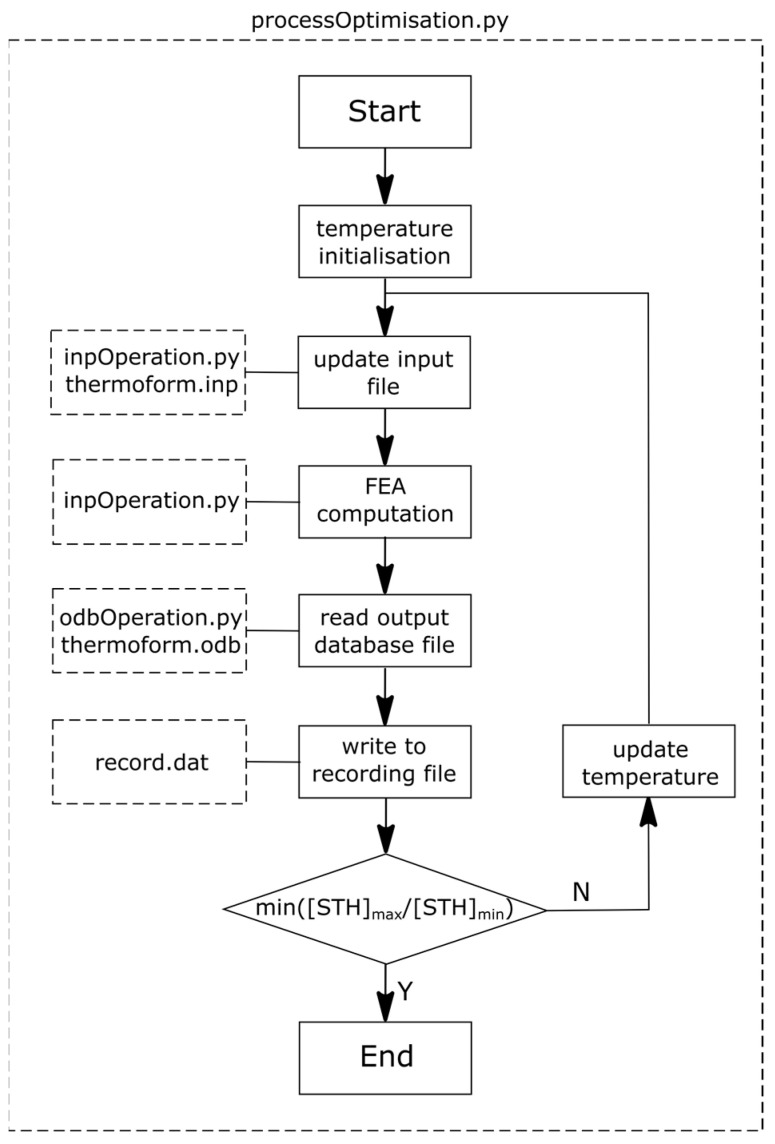
Optimisation flowchart.

**Figure 4 polymers-13-00654-f004:**
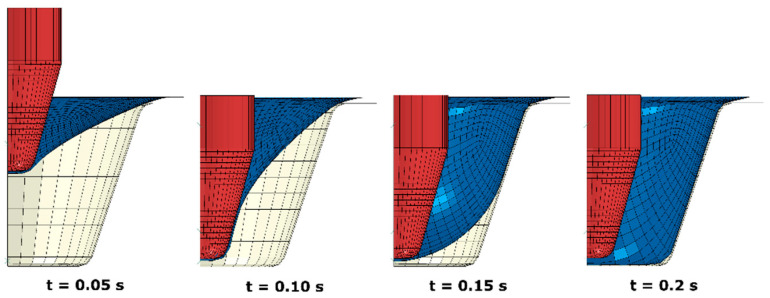
Shape evolution of poly (L-lactic acid) PLA sheet (blue) at forming process (T_p_ = 77 °C).

**Figure 5 polymers-13-00654-f005:**
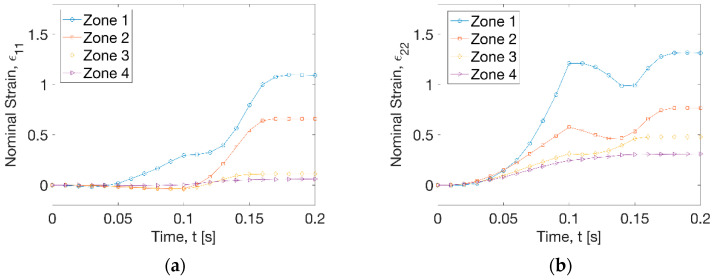
Strain history of 4 zones (T_p_ = 77 °C). (**a**) Hoop strain, (**b**) axial strain.

**Figure 6 polymers-13-00654-f006:**
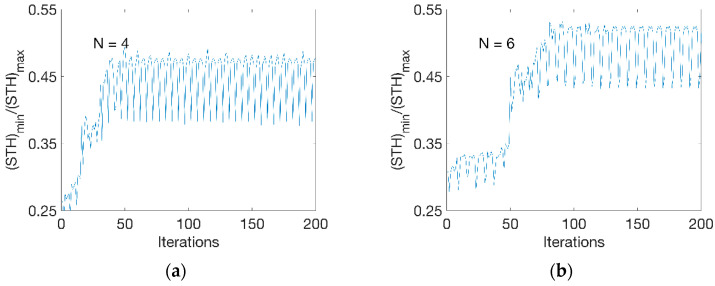
Optimisation process. (**a**) Evolution at N = 4, (**b**) evolution at N = 6.

**Figure 7 polymers-13-00654-f007:**
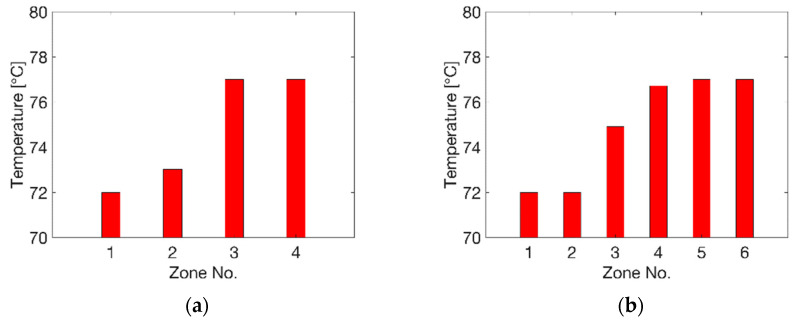
Optimal temperature values of each zone. (**a**) N = 4, (**b**) N = 6.

**Figure 8 polymers-13-00654-f008:**
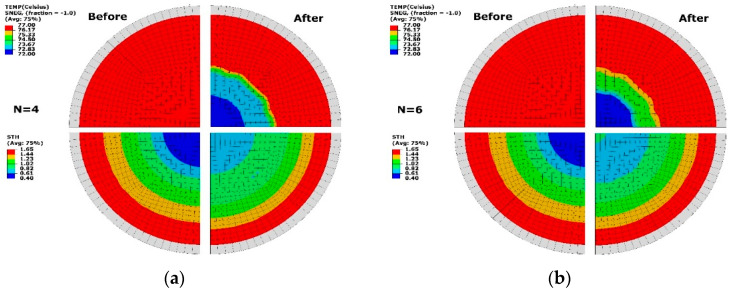
Temperature (TEMP, top) and thickness (STH, bottom) before and after optimisation. (**a**) N = 4, (**b**) N = 6.

**Figure 9 polymers-13-00654-f009:**
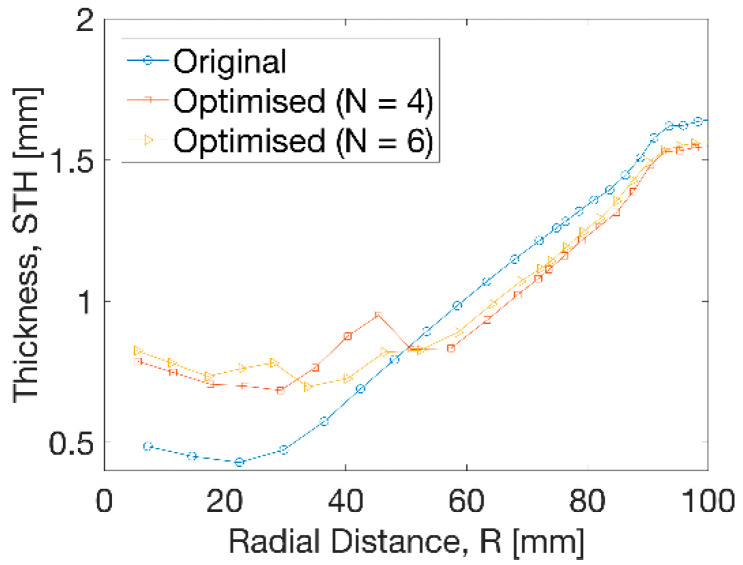
Thickness distribution at different processing conditions.

**Figure 10 polymers-13-00654-f010:**
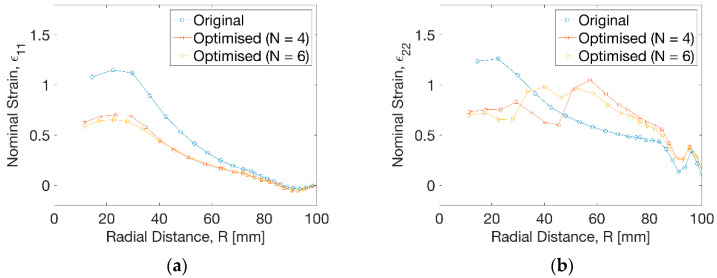
Strain distribution at different processing conditions. (**a**) Hoop strain, (**b**) axial strain.

**Figure 11 polymers-13-00654-f011:**
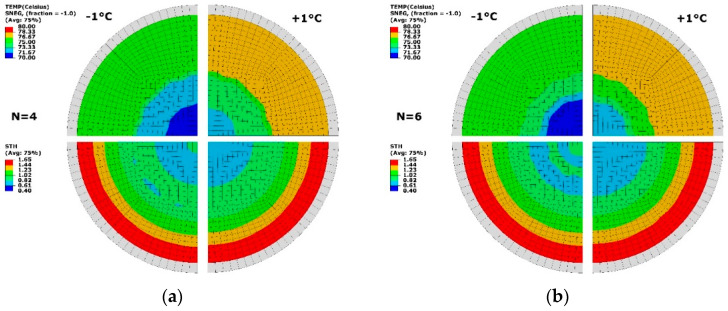
Temperature (TEMP, top) and thickness (STH, bottom) under temperature perturbation. (**a**) N = 4, (**b**) N = 6.

**Figure 12 polymers-13-00654-f012:**
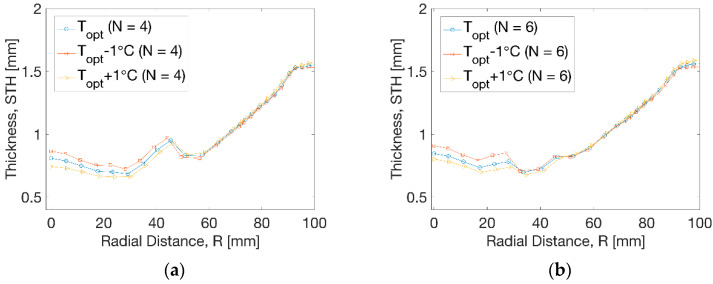
Thickness distribution at different processing conditions. (**a**) N = 4, (**b**) N = 6.

## Data Availability

Data sharing not applicable.

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
