# Peer review of "Optimisation on Thermoforming of Biodegradable Poly (Lactic Acid) (PLA) by Numerical Modelling"

_polymers, 2021, doi:10.3390/polym13040654_

Round 1

Reviewer 1 Report

Dear Authors

This paper is interesting. Results are very well describe. This is worthy to publish.

Below short comments and questions:

  1. Line 94: “Vogel temperature (Tvogel, Tvogel s)” this should probably be corrected
  2. Line 115: the pressure unit should be give in Pa (SI system)
  3. Line 122: “2” change on “two”
  4. Lines 236 and 244: Throughout the text, temperatures are given without decimals. So the accuracy of temperatures should also be without decimals.
  5. On figures 5, 9, 10 12 lines and measurements points are not clear. I propose use colours.

Reviewer 2 Report

My comments are in the file attached

Reviewer 3 Report

Optimisation on thermoforming of biodegradable poly (lactic 2 acid) (PLA) by numerical modelling

Comments:

  1. In Introduction, author described past work, but little comment on the contribution and shortcoming. Author need to provide critical comments.
  2. More applications are required to enhance the introduction part.
  3. Its better, author should provide a comparsion with previously published result to enhance the quality of the presented results.
  4. The new model was derived in the manuscript. However, some limitations were ignored or assumption with the model. Therefore, could the final resolution support the results or application? Or how to assess the deviation?
  5. Is it possible to get a reasonable or optimal composition for the model?
  6. The author should update the manuscript with appropriate and relevant research questions. This would help readers to link what is known in the literature with the novelty of this study. The research questions may be stated at the end of the introduction. However, a new subsection may be raised to pose the research questions.
